# The Level of Processing, Nutritional Composition and Prices of Canadian Packaged Foods and Beverages with and without Gluten-Free Claims

**DOI:** 10.3390/nu13041183

**Published:** 2021-04-02

**Authors:** Laura Vergeer, Beatriz Franco-Arellano, Gabriel B. Tjong, Jodi T. Bernstein, Mary R. L’Abbé

**Affiliations:** 1Department of Nutritional Sciences, Temerty Faculty of Medicine, University of Toronto, Toronto, ON M5S 1A8, Canada; laura.vergeer@mail.utoronto.ca (L.V.); beatriz.francoarellano@ontariotechu.ca (B.F.-A.); gabriel.tjong@mail.utoronto.ca (G.B.T.); jodi.bernstein@mail.utoronto.ca (J.T.B.); 2Faculty of Health Sciences, Ontario Tech University, Oshawa, ON L1G 0C5, Canada

**Keywords:** gluten-free, nutrition claim, nutritional composition, price, food processing

## Abstract

Little is known about the healthfulness and cost of gluten-free (GF) foods, relative to non-GF alternatives, in Canada. This study compared the extent of processing, nutritional composition and prices of Canadian products with and without GF claims. Data were sourced from the University of Toronto Food Label Information Program (FLIP) 2013 (*n* = 15,285) and 2017 (*n* = 17,337) databases. Logistic regression models examined the association of NOVA processing category with GF claims. Calorie/nutrient contents per 100 g (or mL) were compared between GF and non-GF products. Generalized linear models compared adjusted mean prices per 100 g (or mL) of products with and without GF claims. The prevalence of GF claims increased from 7.1% in 2013 to 15.0% in 2017. GF claims appeared on 17.0% of ultra-processed foods, which were more likely to bear GF claims products than less-processed categories. Median calories and sodium were significantly higher in GF products; no significant differences were observed for saturated fat or sugars. Compared to non-GF products, adjusted mean prices of GF products were higher for 10 food categories, lower for six categories and not significantly different for six categories. Overall, GF claims are becoming increasingly prevalent in Canada; however, they are often less healthful and more expensive than non-GF alternatives, disadvantaging consumers following GF diets.

## 1. Introduction

“Gluten-related disorders” is an umbrella-term for several conditions causing adverse reactions to the ingestion of gluten-containing foods [1], including celiac disease, wheat allergies and non-celiac gluten sensitivity [2]. Celiac disease is an autoimmune condition characterized by a heightened immunological response to ingested gluten, believed to affect about 1% of the general population [3]. Although, it is estimated that less than 10% of people with celiac disease have been diagnosed [4]. Wheat allergies and non-celiac gluten sensitivity also affect approximately 1% and 6% of the population, respectively [2,5]. Symptoms of gluten-related disorders may manifest as gastrointestinal, dermatological, and endocrinological problems, among others [2,5]. Following a gluten-free (GF) diet is typically recommended to help alleviate symptoms [2,6].

In Canada, packaged foods and beverages must have <20 ppm of gluten, in order to display a GF claim, the use of which is voluntary [7]. Unlike certain other jurisdictions (e.g., the European Union [8]), there is no regulation in Canada to prevent the use of GF claims on foods that typically do not contain gluten (e.g., fruits and vegetables, dairy products) [7,9]. Products with GF claims are becoming increasingly widespread, despite GF diets being recommended for only a small proportion of the population (i.e., those with gluten-related disorders) [2,3,4]. The market for GF products has experienced rapid annual growth in recent years [10], more so than any other food intolerance category in Canada [11]. Evidence suggests that Canadian consumers often perceive GF products as “healthier” [4], potentially contributing to the growing prevalence of GF claims on packaged foods and beverages. Data from 2007–2013 indicate that GF alternatives are most commonly snacks, bakery products, sauces and seasonings, meat and poultry products; in combination, these food categories represent approximately 60% of all available GF products in Canada [4]. The global market for GF products was worth approximately $5.6 billion USD as of 2020 and is expected to continue growing by 8.1% per year, reaching an estimated value of $8.3 billion USD by 2025 [11].

Although the availability of GF products in the marketplace is expected to continually increase [4], little is known about the prevalence and marketing of GF products in Canada. Previous research has suggested that products with GF claims tend to be more expensive and of comparable or inferior nutritional quality to non-GF alternatives [12,13,14,15,16,17,18,19,20]. However, much of this research is outdated, not specific to the Canadian market, and limited to few food categories with small sample sizes. The purpose of this study was therefore to provide comprehensive and updated estimates of the prevalence, nutritional quality and prices of products carrying GF claims in the Canadian packaged food supply. Specifically, this study aimed to: determine the number and proportion of Canadian packaged foods and beverages carrying a GF claim in 2013 and 2017; examine whether the presence of GF claims was associated with level of processing; and compare the nutritional composition and prices of products with and without GF claims.

## 2. Materials and Methods

### 2.1. Food Composition Data

This study used the University of Toronto Food Label Information Program (FLIP), a database of packaged food and beverage labels, described elsewhere [21,22]. Data were collected from a single outlet of major Canadian grocery retailers in 2013 (Loblaws, Metro, Safeway, Sobeys) and 2017 (Loblaws, Metro, Sobeys). FLIP includes information, such as a product’s: name, brand and manufacturer; Nutrition Facts table (NFt); ingredients list; store of collection; container size; undiscounted price; and photographs of all sides of product packages. Products in FLIP are classified according to the major (*n* = 24) and minor (*n* = 153) food categories in Health Canada’s Table of Reference Amounts for Food (TRA) [23]. Infant and toddler foods and meal replacements were excluded from this study.

### 2.2. Identification of GF Claims

GF claims were identified by systematically reviewing the photographs of product labels in FLIP 2013 (*n* = 15,285) and 2017 (*n* = 17,337). GF claims were coded as present if a gluten-free declaration or symbol appeared on the package. Classification of GF claims was conducted by one researcher and validation of a random 10% of products was conducted by a second researcher. Inter-rater reliability was calculated using Cohen’s Kappa and found almost perfect agreement (kappa = 0.94). Discrepancies were discussed and a final GF classification was agreed upon. 

### 2.3. Level of Processing

The level of processing was determined using the NOVA food processing classification system [24]. NOVA is currently the most frequently and widely used food processing classification system, and has been applied in numerous countries, including Canada [24,25,26]. Products were classified into one of four categories: (1) unprocessed or minimally processed foods; (2) processed culinary ingredients; (3) processed foods and beverages; and (4) ultra-processed products. The level of processing was determined only for products collected in FLIP 2017 (*n* = 17,337). Classification of the entire sample of products was checked twice by the first author and a random 20% of the sample was independently categorized by a second researcher (weighted Cohen’s Kappa = 0.84).

### 2.4. Nutritional Composition

Calories, sodium, saturated fat and total sugars per 100 g or mL (depending on the unit specified for the relevant TRA minor food category) were calculated for products in FLIP 2017. These nutrients of public health concern contribute to poor diet quality, and are associated with the development of obesity and non-communicable diseases (NCDs) [27]. Products missing information for ≥1 nutrient(s) examined were excluded (*n* = 148), resulting in the inclusion of 17,189 products in the nutritional composition analyses. All products were evaluated based on their nutritional composition “as sold” (rather than “as prepared”) to account for variation in preparation instructions between manufacturers, and to facilitate comparisons of these results with those of previous studies examining the healthfulness of foods in relation to nutrition claims [28,29,30,31,32].

### 2.5. Price

Undiscounted price was standardized per 100 g (or mL). Products with a missing price and/or container size data were further excluded (*n* = 964), resulting in a sample size for the price analyses of 16,225 products in FLIP 2017. The type of brand under which a product is offered was included as a covariate in the adjusted price analyses, as it is known to influence food prices [33,34]. Products were classified into one of four brand type categories: (1) multinational (brands offered by major multinational companies; e.g., Kellogg’s *Frosted Flakes*); (2) domestic/other (brands offered by Canadian companies or smaller boutique companies; e.g., *Chapman’s* ice cream); (3) private label premium (brands offered by retailers under a premium label; e.g., Sobeys’ *Sensations by Compliments*); or (4) private label discount (brands offered by retailers under a discount label; e.g., Sobeys’ *S!gnal*).

### 2.6. Statistical Analyses

The number and proportion of products in 2013 and 2017 with a GF claim were determined for the total sample and by food category; significant differences between years were examined using Chi-squared tests. The number and proportion of products with GF claims in 2017 was examined by NOVA category, and binary logistic regression models investigated the association between level of processing (independent variable) and the presence of a GF claim (dependent variable), with adjustment for food category. Median calories, sodium, saturated fat and total sugars per 100 g (or mL) in 2017 products with and without GF claims were calculated overall and by food category. Calorie and nutrient amounts in products with and without GF claims were compared using Mann-Whitney U tests. Generalized linear models (GLM), stratified by TRA major food category, compared prices per 100 g or mL (dependent variable) for products in 2017 with and without GF claims (independent variable). Models adjusted for TRA minor food category, container size (in g or mL), brand type (multinational, domestic/other, private label premium, private label discount) and store (Loblaws, Metro, Sobeys) were also constructed. GLM types and link functions were selected according to the distribution of the data within each food category, and Akaike’s Information Criterion (AIC) and the Finite Sample Corrected AIC. The adjusted mean price per 100 g (or mL) of products with and without GF claims was generated using the SPSS EMMEANS function. For all analyses, *p*-values < 0.05 were considered statistically significant. Analyses were conducted using RStudio (version 1.2.5019, RStudio Inc., Boston, MA, USA) and IBM SPSS (version 26.0, IBM Corp., Chicago, IL, USA).

## 3. Results

### 3.1. Prevalence of GF Claims in 2013 and 2017

The number and proportion of packaged foods and beverages with GF claims in 2013 and 2017 are presented in Table 1. Overall, the proportion of packaged foods and beverages with GF claims increased significantly from 7.1% in 2013 (*n* = 1088) to 15.0% in 2017 (*n* = 2601; *p* < 0.001). The prevalence of GF claims varied by food category, with the greatest proportion observed for dessert toppings and fillings in 2013 (16.1%) and snacks in 2017 (31.9%). Between 2013 and 2017, the proportion of products with GF claims increased significantly for 18 food categories, and did not change significantly for the 4 remaining categories, with the largest increases observed in salads, eggs and egg substitutes and beverages-food categories that do not normally contain gluten.

### 3.2. Level of Processing

Table 2 displays the proportion of products in 2017 with and without GF claims for each level of processing assessed using the NOVA system. GF claims were present on 17.0% (*n* = 2163) of ultra-processed products, 15.0% (*n* = 82) of processed culinary ingredients, 11.6% (*n* = 256) of unprocessed or minimally processed foods, and 5.4% (*n* = 100) of processed foods. There was considerable variation in the presence of GF claims by food category for the different levels of processing. Processed foods (OR = 0.305, 95% CI = 0.243–0.383, *p* < 0.001), processed culinary ingredients (OR = 0.715, 95% CI = 0.550–0.930, *p* = 0.01) and unprocessed/minimally processed foods (OR = 0.648, 95% CI = 0.541–0.776, *p* < 0.001) were less likely to bear GF claims than ultra-processed products.

### 3.3. Nutritional Composition

Median (±IQR) calorie, sodium, saturated fat and total sugars contents per 100 g (or mL) for products in 2017 with and without GF claims are shown in Table 3, overall and by food category.

#### 3.3.1. Calories

Median calorie contents per 100 g (or mL) were significantly higher for products with GF claims than without (300 ± 303 kcal vs. 255 ± 282 kcal, *p* < 0.001) among the total sample, and for 5 food categories (bakery products; cereals and other grain products; dessert toppings and fillings; fruit and fruit juices; and meat, poultry, their products and substitutes). For 10 food categories, products with GF claims had a significantly lower median calorie content per 100 g (or mL) than products without GF claims: dairy products and substitutes; desserts; eggs and egg substitutes; fats and oils; miscellaneous category; combination dishes; nuts and seeds; potatoes, sweet potatoes and yams; snacks; and soups.

#### 3.3.2. Sodium

Compared with products without GF claims, those with GF claims had a significantly higher median sodium content per 100 g (or mL) among the total sample (275 ± 563 mg vs. 264 ± 515 mg, *p* = 0.002) and for 8 food categories: dessert toppings; eggs and egg substitutes; fats and oils; meat, poultry, their products and substitutes; miscellaneous category, nuts and seeds; snacks; and vegetables. Products with GF claims had a lower median sodium amount than those without GF claims for 7 categories: bakery products; dairy products and substitutes; desserts; fruit and fruit juices; combination dishes; sauces, dips, gravies and condiments; and soups.

#### 3.3.3. Saturated Fat

Median saturated fat amounts per 100 g (or mL) of products with and without GF claims did not differ significantly among the total sample (1 ± 4 g vs. 5 ± 16 g, *p* = 0.70). Compared to products without GF claims, those with GF claims had median saturated fat contents that were significantly higher for 3 food categories (bakery products; dessert toppings and fillings; meat, poultry, their products and substitutes) and lower for 6 categories (dairy products and substitutes; desserts; eggs and egg substitutes; fats and oils; combination dishes; and snacks). There were significant differences in the distribution of saturated fat contents per 100 g (or mL) for an additional 5 food categories; however, median saturated fat contents were 0 g for products with and without GF claims. 

#### 3.3.4. Total Sugars

Overall, products with and without GF claims did not differ significantly in terms of median total sugars content per 100 g or mL (4 ± 18 g vs. 5 ± 16 g, *p* = 0.14). Compared to products without GF claims, the median total sugars content of products with GF claims was significantly higher for 6 food categories (bakery products; dairy products; dessert toppings and fillings; fruit and fruit juices; potatoes, sweet potatoes and yams; and vegetables) and lower for 4 categories (cereals and other grain products; desserts; meat, poultry, their products and substitutes; and snacks).

### 3.4. Price

Unadjusted and adjusted mean (±SE) prices per 100 g (or mL) of products in 2017 with and without GF claims are presented by food category in Table 4. After adjustment for container size, store, brand type and food category, the mean price of products with GF claims was greater than that of products without GF claims for 10 food categories: bakery products ($2.26 ± 0.07 vs. $1.46 ± 0.02, *p* < 0.001); cereals and other grain products ($1.27 ± 0.08 vs. $0.96 ± 0.05, *p* < 0.001); fats and oils ($1.47 ± 0.09 vs. $1.17 ± 0.04, *p* = 0.001); fruit and fruit juices ($1.05 ± 0.09 vs. $0.74 ± 0.05, *p* < 0.001); miscellaneous category ($2.40 ± 0.16 vs. $1.81 ± 0.10, *p* < 0.001); combination dishes ($1.62 ± 0.07 vs. $1.30 ± 0.03, *p* < 0.001); nuts and seeds ($2.17 ± 0.25 vs. $1.66 ± 0.16, *p* = 0.002); potatoes, sweet potatoes and yams ($1.11 ± 0.22 vs. $0.52 ± 0.03, *p* = 0.01); snacks ($2.85 ± 0.14 vs. $1.97 ± 0.08, *p* < 0.001); and soups ($1.33 ± 0.06 vs. $1.15 ± 0.04, *p* < 0.001). For 6 categories, products with GF claims had a lower mean adjusted price than products without GF claims: beverages ($0.81 ± 0.15 vs. $1.43 ± 0.17, *p* < 0.001); dairy products and substitutes ($0.93 ± 0.03 vs. $1.06 ± 0.02, *p* < 0.001); desserts ($0.59 ± 0.04 vs. $0.75 ± 0.03, *p* < 0.001); dessert toppings and fillings ($0.50 ± 0.05 vs. $1.31 ± 0.07, *p* < 0.001); eggs and egg substitutes ($0.45 ± 0.07 vs. $0.78 ± 0.07, *p* = 0.004); and marine and fresh water animals ($2.24 ± 0.28 vs. $2.91 ± 0.14, *p* = 0.01). No significant differences in adjusted mean prices were observed for the remaining 6 food categories.

## 4. Discussion

This study provides a comprehensive analysis of the prevalence, healthfulness and price of products with GF claims in Canada. GF claims were increasingly common in the Canadian packaged food supply, were most likely to appear on ultra-processed foods, and tended to be higher in calories, nutrients of public health concern, and price per 100 g (or mL) than non-GF alternatives within several food categories. This trend was, however, not consistent across all food categories, with several instances where products with GF claims were lower, or not significantly different, in energy and nutrient density or price, than those without.

Products with GF claims were most commonly displayed on ultra-processed foods and beverages, excess consumption of which has been associated with poor diet quality, obesity and NCDs [25,35]. Ultra-processed products in Canada are typically energy-dense and high in nutrients of public health concern [26], which aligns with our finding that GF products had significantly higher median calorie, sodium, saturated fat and/or sugars contents than non-GF products for several of the food categories examined. Products with GF claims were also higher in price per 100 g (or mL) for 10 of these 22 food categories. The results from this study are therefore consistent with previous research suggesting that GF products tend to be more expensive and of poorer or comparable nutritional quality to non-GF alternatives [12,13,14,15,16,17,18,19,20,36,37,38,39,40,41,42]. However, the nutritional composition and prices of products with GF claims relative to those without varies by food category, as evidenced by the findings of this work and previous studies [36,37,42,43,44].

These findings have implications for both consumers required to follow a GF diet for medical conditions, and those electing to do so for lifestyle reasons. Consumers with diagnosed gluten-related disorders may not be able to consume a nutritious diet from the available GF packaged foods, many of which were found to be energy-dense and high in nutrients of public health concern. Previous research also indicates that products with GF claims tend to be low in protein and fiber [15]. Nonetheless, many healthier, less-processed foods and beverages are naturally GF, such as most dairy products, fruits and vegetables, nuts and seeds, legumes, and vegetable oils. The fact that these categories still bear GF claims, as evidenced by our results, suggests that GF labelling is sometimes used as a form of marketing to make GF products appear healthier than their non-GF counterparts. Evidence suggests nearly one-third of Canadians (~10 million) look for GF products when grocery shopping, with approximately seven million choosing GF products because they perceive them as healthier [4]. Findings from this study and others indicate that consuming packaged foods and beverages with GF claims in the absence of gluten-related disorders offers no nutritional benefits over products without GF claims [13,14,15,20].

The higher cost of products with GF claims, relative to those without such claims, may also disadvantage consumers of GF foods, irrespective of their reasons for choosing these products. For consumers required to consume a GF diet, the price differential observed within several food categories may have negative financial and health implications, particularly for those of lower socioeconomic status. Given that cost is one of the strongest determinants of food selection in Canada [45], consumers with gluten-related disorders may be forced to choose non-GF products if they are priced significantly lower than non-GF alternatives, with potentially harmful impacts on health. Importantly, this study found that products with GF claims tended to be more expensive than non-GF products in several ‘core’ food categories, such as bakery products, cereals and other grain products, fruits and fruit juices, and nuts and seeds, among others. Similarly, for several food categories that typically contain gluten, the adjusted mean prices of products with GF claims were significantly higher than those without (e.g., bakery products, cereals and other grain products, combination dishes, snacks). The higher cost of many GF foods, relative to non-GF alternatives, indicates that consumers choosing GF foods for purported health benefits risk paying more for a GF product that may actually be less healthy than a gluten-containing version of the product (or a product that is naturally GF, but not marketed as such).

Although the use of GF claims is more tightly regulated in Canada than many other countries [4], additional nutrition labelling interventions to support consumers in making healthier food choices are warranted. In Canada, consumers often neglect to use, or have difficulty interpreting, the NFt, and are easily influenced by the use of nutrition claims promoting positive aspects of foods [46,47], regardless of whether the food is considered a healthy option. Interpretative, salient front-of-package (FOP) labelling on all packaged foods may enable consumers to more easily and accurately assess the nutritional quality of products with GF claims, relative to those without. Mandatory FOP symbols on Canadian foods high in sodium, saturated fat and/or total sugars were proposed in 2018 [48], but not yet implemented. 

This study is strengthened by the use of a large, highly representative dataset of packaged foods and beverage products in Canada available in 2013 and 2017. Access to branded food composition data enabled a more accurate examination of the prevalence, processing level, nutritional quality and price of products without and without GF claims, compared with studies using generic food composition databases (e.g., the Canadian Nutrient File), which do not include data on the presence of claims on products [49,50]. In addition, the price analyses accounted for important variables that are known to influence food prices but that have been largely unaccounted for in previous research (i.e., container size, store, brand type, food category) [33,34,51,52]. Nonetheless, this study is not without limitations. First, given that FLIP data were collected from only a few Canadian grocery retail outlets at single points in time, this sample does not necessarily capture all products with GF claims available in Canada. Similarly, it does not account for price differences between geographic locations, identical products offered by different stores or at various time points, or price promotions. Future research is needed to monitor trends in the use of GF claims in the Canadian market over time, including how products with GF claims are marketed to consumers (i.e., through the display of other nutrition claims, front-of-package labelling, pricing, advertising), and the extent to which marketing affects consumer purchasing behaviours.

## 5. Conclusions

In Canada, ~15% of the packaged food supply carried a GF claim as of 2017, a doubling from 7.1% in 2013. As of 2017, GF claims most commonly appeared on ultra-processed foods, and for several food categories, products with GF claims were higher in calories, sodium, saturated fat, sugars and/or price, compared to products without GF claims. However, for other categories, foods with GF claims were lower or comparable in energy, nutrients of concern and price than products without these claims. Overall, consumers requiring or preferring GF foods may be disadvantaged if certain types of products bearing GF claims are less healthy and more expensive than comparable products without such claims.

## Figures and Tables

**Table 1 nutrients-13-01183-t001:** The number and proportion of Canadian packaged foods and beverages with gluten-free (GF) claims in 2013 and 2017.

Food Category ^1^	2013			2017			*p*-Value ^2^
	Total	Products with GF Claim		Total	Products with GF Claim		
	*n*	*n*	%	*n*	*n*	%	
A. Bakery products	2097	150	7.2	2775	375	13.5	**<0.001**
B. Beverages	482	13	2.7	852	77	9.0	**<0.001**
C. Cereals and other grain products	1126	109	9.7	1276	208	16.3	**<0.001**
D. Dairy products and substitutes	1225	49	4.0	1498	170	11.3	**<0.001**
E. Desserts	829	46	5.5	679	89	13.1	**<0.001**
F. Dessert toppings and fillings	118	19	16.1	94	13	13.8	0.79
G. Eggs and egg substitutes	56	1	1.8	61	5	8.2	0.25
H. Fats and oils	537	54	10.1	656	100	15.2	**0.01**
I. Marine and fresh water animals	442	8	1.8	446	13	2.9	0.39
J. Fruit and fruit juices	1078	26	2.4	1061	81	7.6	**<0.001**
K. Legumes	182	7	3.8	188	21	11.2	**0.01**
L. Meat, poultry, their products and substitutes	910	138	15.2	962	270	28.1	**<0.001**
M. Miscellaneous category	476	37	7.8	558	118	21.1	**<0.001**
N. Combination dishes	1231	35	2.8	1139	99	8.7	**<0.001**
O. Nuts and seeds	202	25	12.4	255	67	26.3	**<0.001**
P. Potatoes, sweet potatoes and yams	140	1	0.7	132	6	4.5	0.11
Q. Salads	70	1	1.4	130	17	13.1	**0.01**
R. Sauces, dips, gravies and condiments	1246	134	10.8	1250	258	20.6	**<0.001**
S. Snacks	746	117	15.7	865	276	31.9	**<0.001**
T. Soups	457	39	8.5	480	82	17.1	**<0.001**
U. Sugars and sweets	796	46	5.8	1109	189	17.0	**<0.001**
V. Vegetables	839	33	3.9	871	67	7.7	**0.001**
TOTAL	15,285	1088	7.1	17,337	2601	15.0	**<0.001**

^1^ Food categories are defined in Health Canada’s Table of Reference Amounts for Food [23]. ^2^ Based on Chi-squared tests to compare the proportion of products that displayed GF claims in FLIP 2013 and 2017; *p*-values < 0.05 were considered statistically significant and are shown in boldface.

**Table 2 nutrients-13-01183-t002:** The prevalence of products with gluten-free (GF) claims in 2017 by NOVA processing category.

Food Category ^1^	Ultra-Processed Products			Processed Foods and Beverages			Processed Culinary Ingredients			Unprocessed/Minimally Processed		
	*n*	GF (*n*)	GF (%)	*n*	GF (*n*)	GF (%)	*n*	GF (*n*)	GF (%)	*n*	GF (*n*)	GF (%)
A. Bakery products	2628	375	14.3	146	0	0.0	0	0	0.0	1	0	0.0
B. Beverages	743	65	8.7	6	0	0.0	0	0	0.0	103	12	11.7
C. Cereals and other grain products	511	67	13.1	23	2	8.7	8	4	50.0	734	135	18.4
D. Dairy products and substitutes	947	158	16.7	407	0	0.0	0	0	0.0	144	12	8.3
E. Desserts	679	89	13.1	0	0	0.0	0	0	0.0	0	0	0.0
F. Dessert toppings and fillings	94	13	13.8	0	0	0.0	0	0	0.0	0	0	0.0
G. Eggs and egg substitutes	9	5	55.6	0	0	0.0	0	0	0.0	52	0	0.0
H. Fats and oils	425	89	20.9	0	0	0.0	231	11	4.8	0	0	0.0
I. Marine and fresh water animals	187	12	6.4	202	1	0.5	0	0	0.0	57	0	0.0
J. Fruit and fruit juices	304	12	3.9	213	16	7.5	0	0	0.0	544	53	9.7
K. Legumes	5	4	80.0	91	13	14.3	0	0	0.0	92	4	4.3
L. Meat, poultry, their products and substitutes	838	258	30.8	93	10	10.8	0	0	0.0	31	2	6.5
M. Miscellaneous category	424	77	18.2	0	0	0.0	104	35	33.7	30	6	20.0
N. Combination dishes	1139	99	8.7	0	0	0.0	0	0	0.0	0	0	0.0
O. Nuts and seeds	59	25	42.4	0	0	0.0	48	22	45.8	148	20	13.5
P. Potatoes, sweet potatoes and yams	103	2	1.9	15	2	13.3	0	0	0.0	14	2	14.3
Q. Salads	114	14	12.3	14	2	14.3	0	0	0.0	2	1	50.0
R. Sauces, dips, gravies and condiments	1213	258	21.3	0	0	0.0	37	0	0.0	0	0	0.0
S. Snacks	723	261	36.1	132	14	10.6	0	0	0.0	10	1	10.0
T. Soups	480	82	17.1	0	0	0.0	0	0	0.0	0	0	0.0
U. Sugars and sweets	984	179	18.2	0	0	0.0	119	10	8.4	6	0	0.0
V. Vegetables	144	19	13.2	493	40	8.1	0	0	0.0	234	8	3.4
TOTAL	12,753	2163	17.0	1835	100	5.4	547	82	15.0	2202	256	11.6

^1^ Food categories are defined in Health Canada’s Table of Reference Amounts for Food [23].

**Table 3 nutrients-13-01183-t003:** The nutritional composition per 100 g (or mL) of products with and without gluten-free (GF) claims in 2017.

Food Category ^1^	Number of Products (*n*)			Calories per 100 g (or mL) (kcal)			Sodium per 100 g (or mL) (mg)			Saturated Fat per 100 g (or mL) (g)			Total Sugars per 100 g (or mL) (g)		
				GF Claim	No GF Claim	*p*-Value ^2^	GF Claim	No GF Claim	*p*-Value ^2^	GF Claim	No GF Claim	*p*-Value ^2^	GF Claim	No GF Claim	*p*-Value ^2^
	Total	GF Claim	No GF Claim	Median (IQR)	Median (IQR)		Median (IQR)	Median (IQR)		Median (IQR)	Median (IQR)		Median (IQR)	Median (IQR)	
A. Bakery products	2770	377	2393	400 (90)	393 (161)	**<0.001**	280 (303)	355 (267)	**<0.001**	4 (6)	3 (7)	**0.02**	20 (24)	15 (25)	**0.02**
B. Beverages	835	75	760	19 (14)	22 (45)	0.94	0 (23)	6 (20)	0.07	0 (0)	0 (0)	0.55	3 (5)	5 (11)	0.10
C. Cereals and other grain products	1274	208	1066	367 (33)	356 (25)	**<0.001**	10 (122)	12 (291)	0.12	0 (1)	0 (1)	0.90	0 (7)	2 (7)	**<0.001**
D. Dairy products and substitutes	1495	170	1325	57 (61)	240 (279)	**<0.001**	45 (28)	350 (651)	**<0.001**	0 (2)	10 (15)	**<0.001**	4 (9)	3 (10)	**0.003**
E. Desserts	679	89	590	100 (33)	144 (138)	**<0.001**	32 (49)	56 (55)	**<0.001**	2 (3)	3 (5)	**0.04**	11 (5)	16 (9)	**<0.001**
F. Dessert toppings and fillings	94	13	81	400 (29)	308 (211)	**<0.001**	200 (58)	87 (210)	**0.01**	7 (3)	0 (2)	**<0.001**	57 (5)	47 (41)	**0.04**
G. Eggs and egg substitutes	61	5	56	56 (79)	132 (6)	**0.004**	302 (32)	127 (8)	**<0.001**	0 (2)	3 (0)	**0.04**	0 (0)	0 (0)	N/A
H. Fats and oils	656	100	556	400 (400)	600 (467)	**<0.001**	600 (433)	550 (767)	**0.03**	3 (5)	7 (12)	**<0.001**	0 (7)	0 (7)	0.42
I. Marine and fresh water animals	446	13	433	100 (105)	140 (103)	0.54	425 (164)	360 (233)	0.67	1 (1)	1 (2)	0.21	1 (2)	0 (2)	0.24
J. Fruit and fruit juices	1053	78	975	65 (277)	50 (28)	**<0.001**	3 (8)	6 (12)	**0.007**	0 (0)	0 (0)	**<0.001**	13 (38)	11 (6)	**0.005**
K. Legumes	188	21	167	104 (63)	106 (260)	0.14	13 (279)	12 (86)	0.19	0 (0)	0 (0)	0.25	1 (6)	1 (2)	0.92
L. Meat, poultry, their products and substitutes	959	268	691	233 (166)	211 (111)	**0.04**	800 (473)	553 (432)	**<0.001**	5 (8)	3 (6)	**0.03**	0 (2)	1 (2)	**0.002**
M. Miscellaneous category	549	114	435	324 (364)	367 (122)	**<0.001**	988 (7422)	684 (3819)	**0.01**	0 (0)	0 (3)	**<0.001**	2 (33)	10 (40)	0.08
N. Combination dishes	1129	99	1030	133 (115)	198 (124)	**<0.001**	276 (172)	387 (249)	**<0.001**	1 (2)	2 (3)	**0.004**	3 (2)	3 (3)	0.92
O. Nuts and seeds	251	63	188	600 (70)	633 (69)	**<0.001**	31 (219)	0 (28)	**<0.001**	8 (4)	7 (4)	0.09	6 (11)	4 (4)	0.21
P. Potatoes, sweet potatoes and yams	132	6	126	74 (9)	153 (68)	**0.002**	156 (137)	235 (280)	0.05	0 (0)	0 (1)	**0.003**	2 (0)	0 (2)	**0.02**
Q. Salads	129	16	113	152 (55)	140 (80)	0.49	200 (169)	288 (240)	0.36	2 (2)	2 (1)	0.53	4 (5)	3 (3)	0.39
R. Sauces, dips, gravies and condiments	1240	258	982	133 (159)	125 (136)	0.19	446 (581)	625 (1067)	**<0.001**	0 (2)	0 (1)	**0.001**	5 (16)	6 (18)	0.47
S. Snacks	854	268	586	500 (65)	520 (100)	**<0.001**	520 (413)	480 (490)	**0.005**	3 (2)	5 (5)	**<0.001**	3 (6)	4 (4)	**0.001**
T. Soups	480	82	398	56 (102)	80 (285)	**<0.001**	251 (182)	528 (1500)	**<0.001**	0 (1)	0 (2)	**0.003**	1 (2)	2 (4)	0.07
U. Sugars and sweets	1055	179	876	357 (182)	380 (182)	0.10	27 (67)	36 (88)	0.20	0 (17)	0 (17)	0.47	53 (22)	53 (23)	0.31
V. Vegetables	860	67	793	28 (58)	33 (51)	0.60	284 (836)	152 (571)	**<0.001**	0 (0)	0 (0)	0.28	3 (4)	2 (4)	**0.04**
TOTAL	17,189	2569	14,620	300 (303)	255 (282)	**<0.001**	275 (563)	264 (515)	**0.002**	1 (4)	1 (5)	0.70	4 (18)	5 (16)	0.14

^1^ Food categories are defined in Health Canada’s Table of Reference Amounts for Food [23]. ^2^ Based on Mann Whitney U tests to compare amounts of calories or the nutrient of interest in products with versus without a GF claim; *p*-values < 0.05 were considered statistically significant and are shown in boldface.

**Table 4 nutrients-13-01183-t004:** The mean price per 100 g (or mL) of products with and without gluten-free (GF) claims in 2017.

Food Category ^1^	No. of Products (*n*)	GF Claim (*n*)	No GF Claim (*n*)	Mean (SE) Price—GF Claim ($/100 g)	Mean (SE) Price—No GF Claim ($/100 g)	*p*-Value ^2^	Adjusted Mean (SE) Price—GF Claim ($/100 g)^3^	Adjusted Mean (SE) Price—No GF Claim ($/100 g) ^3^	*p*-Value ^2^
A. Bakery products	2644	365	2279	3.81 (0.18)	1.62 (0.02)	**<0.001**	2.26 (0.07)	1.46 (0.02)	**<0.001**
B. Beverages	791	70	721	1.04 (0.17)	2.15 (0.11)	**<0.001**	0.81 (0.15)	1.43 (0.17)	**<0.001**
C. Cereals and other grain products	1242	197	1045	1.62 (0.08)	0.97 (0.02)	**<0.001**	1.27 (0.08)	0.96 (0.05)	**<0.001**
D. Dairy products and substitutes	1405	160	1245	0.86 (0.05)	2.02 (0.05)	**<0.001**	0.93 (0.03)	1.06 (0.02)	**<0.001**
E. Desserts	641	82	559	0.81 (0.07)	1.20 (0.04)	**<0.001**	0.59 (0.04)	0.75 (0.03)	**<0.001**
F. Dessert toppings and fillings	88	13	75	0.78 (0.09)	1.56 (0.11)	**<0.001**	0.50 (0.05)	1.31 (0.07)	**<0.001**
G. Eggs and egg substitutes	54	5	49	0.64 (0.08)	0.81 (0.04)	0.06	0.45 (0.07)	0.78 (0.07)	**0.004**
H. Fats and oils	612	88	524	1.77 (0.16)	1.36 (0.05)	**0.02**	1.47 (0.09)	1.17 (0.04)	**0.001**
I. Marine and fresh water animals	432	13	419	2.35 (0.35)	2.85 (0.08)	0.16	2.24 (0.28)	2.91 (0.14)	**0.01**
J. Fruit and fruit juices	992	72	920	1.85 (0.18)	0.69 (0.02)	**<0.001**	1.05 (0.09)	0.74 (0.05)	**<0.001**
K. Legumes	178	20	158	0.75 (0.12)	0.44 (0.02)	**0.008**	0.68 (0.09)	0.66 (0.07)	0.80
L. Meat, poultry, their products and substitutes	859	242	617	2.74 (0.11)	2.15 (0.05)	**<0.001**	1.91 (0.07)	1.82 (0.05)	0.09
M. Miscellaneous category	529	111	418	4.40 (0.39)	2.98 (0.14)	**0.001**	2.40 (0.16)	1.81 (0.10)	**<0.001**
N. Combination dishes	1068	96	972	1.78 (0.09)	1.43 (0.02)	**<0.001**	1.62 (0.07)	1.30 (0.03)	**<0.001**
O. Nuts and seeds	234	56	178	2.91 (0.25)	2.85 (0.14)	0.83	2.17 (0.25)	1.66 (0.16)	**0.002**
P. Potatoes, sweet potatoes and yams	123	6	117	0.89 (0.23)	0.76 (0.04)	0.56	1.11 (0.22)	0.52 (0.03)	**0.01**
Q. Salads	123	12	111	2.04 (0.22)	1.90 (0.07)	0.57	0.98 (0.48)	1.03 (0.50)	0.45
R. Sauces, dips, gravies and condiments	1189	246	983	1.74 (0.10)	1.76 (0.05)	0.82	1.15 (0.06)	1.13 (0.03)	0.75
S. Snacks	783	241	542	3.05 (0.13)	1.88 (0.04)	**<0.001**	2.85 (0.14)	1.97 (0.08)	**<0.001**
T. Soups	479	82	397	1.95 (0.25)	1.22 (0.06)	**0.005**	1.33 (0.06)	1.15 (0.04)	**<0.001**
U. Sugars and sweets	965	157	808	2.66 (0.13)	2.30 (0.05)	**0.01**	1.52 (0.09)	1.57 (0.08)	0.46
V. Vegetables	794	62	732	3.00 (0.38)	1.11 (0.04)	**<0.001**	0.86 (0.07)	0.85 (0.04)	0.91

^1^ Food categories are defined in Health Canada’s Table of Reference Amounts for Food [23]. ^2^
*p*-value derived from unadjusted or adjusted generalized linear model (GLM); *p*-values < 0.05 were considered statistically significant and are shown in boldface. ^3^ Based on GLM adjusted for TRA minor food category, brand type (multinational, domestic/other, private label premium, private label discount), store (Loblaws, Metro, Sobeys) and container size (in g or mL). Abbreviations: SE = Standard Error.

## Data Availability

The data presented in this study are available on request from the corresponding author.

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
