# Peer review of "The Level of Processing, Nutritional Composition and Prices of Canadian Packaged Foods and Beverages with and without Gluten-Free Claims"

_nutrients, 2021, doi:10.3390/nu13041183_

Round 1
Reviewer 1 Report
The context should be better described. Lines 28-38 and 39-51 should be better described and enlarged. The aim should be clarified.
The authors should be mark the importace to use National Food Composition Databases including processed foods and complex food matricies and report some related references suvh as Durazzo et al. Nutritional composition and antioxidant properties of traditional Italian dishes. Food Chem. 2017 Mar 1 ;218:70-77. doi: 10.1016/j.foodchem.2016.08.120
The subparagraphs 2.2. Identification of GF claims and 2.3. Level of processing should be implemented.
Major details should be given in Table 1.
Tables 2, 3 and 4 should be checked and reritten and better described in the text.
Section Conclusion should be implemented, by inserting limits, advantages, practical applications and future directions.
The linguistic revision of whole manuscript should be carried out.
Author Response
Manuscript ID: nutrients-1156563
Title: The Level of Processing, Nutritional Composition and Prices of Canadian Packaged Foods and Beverages with and without Gluten-Free Claims
Authors: Laura Vergeer, Beatriz Franco-Arellano, Gabriel B. Tjong, Jodi T. Bernstein, Mary R. L’Abbé
Reviewer 1 Comments:
Open Review
(x) I would not like to sign my review report
( ) I would like to sign my review report
English language and style
( ) Extensive editing of English language and style required
(x) Moderate English changes required
( ) English language and style are fine/minor spell check required
( ) I don't feel qualified to judge about the English language and style
Yes |
Can be improved |
Must be improved |
Not applicable |
|
Does the introduction provide sufficient background and include all relevant references? |
( ) |
(x) |
( ) |
( ) |
Is the research design appropriate? |
( ) |
(x) |
( ) |
( ) |
Are the methods adequately described? |
( ) |
(x) |
( ) |
( ) |
Are the results clearly presented? |
( ) |
(x) |
( ) |
( ) |
Are the conclusions supported by the results? |
( ) |
(x) |
( ) |
( ) |
Comments and Suggestions for Authors
We thank Reviewer 1 for taking the time to provide feedback on our manuscript. Please see our responses below. Also, please note that the line numbers refer to those displayed when “Track Changes” is turned on (causing the line numbering to start at 45); different numbers may be visible if this function is disabled.
Comment 1: The context should be better described. Lines 28-38 and 39-51 should be better described and enlarged. The aim should be clarified.
Response 1: Thank you for this comment; however, we feel that the Introduction as written covers all of the relevant context and it is within the typical text length for an Introduction section, consistent with the Nutrients journal format. The first paragraph of the Introduction describes the clinical reasons why people consumer gluten-free products, while the second paragraph describes the gluten-free market in Canada. The third paragraph summarizes existing literature concerning the healthfulness and cost of gluten-free products and highlights limitations of this research to provide a rationale for our study objective, which is described in lines 104-110 (page 2). We believe this provides the background information and clear aim needed for our study.
Comment 2: The authors should be mark the importace to use National Food Composition Databases including processed foods and complex food matricies and report some related references suvh as Durazzo et al. Nutritional composition and antioxidant properties of traditional Italian dishes. Food Chem. 2017 Mar 1 ;218:70-77. doi: 10.1016/j.foodchem.2016.08.120
Response 2: There is currently no national branded food composition database in Canada; however, the FLIP database is the largest branded food database in the country. We have described the use of a large branded food database as a strength of this study in lines 353-358 (page 12). In addition, the national generic food composition database, the Canadian Nutrient File1, is built on food composites and does not include information about claims displayed on food packaging. Therefore, the use of a branded food composition database, like FLIP, is more appropriate. We have added a reference to this section of the Discussion (reference #38; lines 357-358, page 12) that includes a global review of existing branded and generic food composition databases, and describes the limitations of generic databases compared to those with branded information. We believe this reference is more relevant than the Durazzo et al. study suggested by the reviewer.
1Health Canada. Canadian Nutrient File (CNF) – Search by food. Available online: https://food-nutrition.canada.ca/cnf-fce/index-eng.jsp (accessed 23 March 2021).
Comment 3: The subparagraphs 2.2. Identification of GF claims and 2.3. Level of processing should be implemented.
Response 3: Subparagraphs 2.2 and 2.3 are included in lines 122 and 130, respectively (page 2).
Comment 4: Major details should be given in Table 1. Tables 2, 3 and 4 should be checked and reritten and better described in the text.
Response 4: This paper is descriptive in nature and each objective is presented in Tables 1-4. The formatting of Tables 1-4 has been checked and adjusted, where needed, to ensure clarity. We have also checked that they are clearly described in the text.
Comment 5: Section Conclusion should be implemented, by inserting limits, advantages, practical applications and future directions.
Response 5: A “Conclusions” paragraph is included in lines 371-380 (pages 12-13). This section summarizes the main findings and implications of the study. The limitations, strengths and future research directions are described in the preceding paragraph within the Discussion section (page 12, lines 353-370) to be consistent with the Nutrients manuscript format.
Comment 6: The linguistic revision of whole manuscript should be carried out.
Response 6: The manuscript was written by a native English speaker. In addition, three of the co-authors are also native English speakers and have reviewed the manuscript for language accuracy. We have checked it for proper grammar, spelling, punctuation and style.

Reviewer 2 Report
Review of Manuscript Number: ID: Nutrients-1156563 peer-review-v1
Title:“ The Level of Processing, Nutritional Composition and Prices of Canadian Packaged Foods and Beverages with and without Gluten-Free Claims”
Journal: Nutrients
The manuscript deals with an important topic, ie the comparison of the health benefits and prices of gluten-free food and non-gluten-free food. Gluten-free food is an important assortment on the food market, especially for people with celiac disease. This study compared the extent of processing, nutritional composition and price of Canadian products with and without GF claims. Data were sourced from the University of Toronto FLIP 2013 (n=15,285) and 2017 (n=17,337) databases. Median calories and sodium were significantly higher in GF products; no significant differences were observed for saturated fat or sugars. Compared to non-GF products, adjusted mean prices of GF products were higher for 10 food categories, lower for 6 categories and not significantly different for 6 categories. Overall, GF claims are becoming increasingly prevalent in Canada; however, they are often less healthful and more expensive than non-GF alternatives, disadvantaging consumers following GF diets.
I suggest minor editorial changes:
- In Table 3, instead of g / mL, 100g / mL should be g mL-1, 100g mL-1
- in the list of bibliographies, a standardized way of writing literature items consistent with editorial requirements should be used.
Author Response
Manuscript ID: nutrients-1156563
Title: The Level of Processing, Nutritional Composition and Prices of Canadian Packaged Foods and Beverages with and without Gluten-Free Claims
Authors: Laura Vergeer, Beatriz Franco-Arellano, Gabriel B. Tjong, Jodi T. Bernstein, Mary R. L’Abbé
Reviewer 2 Comments
Open Review
(x) I would not like to sign my review report
( ) I would like to sign my review report
English language and style
( ) Extensive editing of English language and style required
( ) Moderate English changes required
( ) English language and style are fine/minor spell check required
(x) I don't feel qualified to judge about the English language and style
Yes |
Can be improved |
Must be improved |
Not applicable |
|
Does the introduction provide sufficient background and include all relevant references? |
(x) |
( ) |
( ) |
( ) |
Is the research design appropriate? |
( ) |
( ) |
( ) |
( ) |
Are the methods adequately described? |
(x) |
( ) |
( ) |
( ) |
Are the results clearly presented? |
(x) |
( ) |
( ) |
( ) |
Are the conclusions supported by the results? |
(x) |
( ) |
( ) |
( ) |
Comments and Suggestions for Authors
Review of Manuscript Number: ID: Nutrients-1156563 peer-review-v1
Title:“ The Level of Processing, Nutritional Composition and Prices of Canadian Packaged Foods and Beverages with and without Gluten-Free Claims”
Journal: Nutrients
The manuscript deals with an important topic, ie the comparison of the health benefits and prices of gluten-free food and non-gluten-free food. Gluten-free food is an important assortment on the food market, especially for people with celiac disease. This study compared the extent of processing, nutritional composition and price of Canadian products with and without GF claims. Data were sourced from the University of Toronto FLIP 2013 (n=15,285) and 2017 (n=17,337) databases. Median calories and sodium were significantly higher in GF products; no significant differences were observed for saturated fat or sugars. Compared to non-GF products, adjusted mean prices of GF products were higher for 10 food categories, lower for 6 categories and not significantly different for 6 categories. Overall, GF claims are becoming increasingly prevalent in Canada; however, they are often less healthful and more expensive than non-GF alternatives, disadvantaging consumers following GF diets.
I suggest minor editorial changes:
- In Table 3, instead of g / mL, 100g / mL should be g mL-1, 100g mL-1
- in the list of bibliographies, a standardized way of writing literature items consistent with editorial requirements should be used.
We thank Reviewer 2 for taking the time to provide feedback on our manuscript. With regards to Table 3, we used “g / mL” to refer to “grams or millilitres”, not “grams per millilitre” as indicated by the reviewer. This has been clarified in the title of Table 3 and elsewhere in the text, and has been changed to per 100 g (or mL) to avoid any misunderstanding. Additionally, we have checked the formatting of references to ensure they are consistent with the journal editorial requirements.
